# TRAINING DEEP NETWORKS WITH STOCHASTIC GRADIENT NORMALIZED BY LAYERWISE ADAPTIVE SECOND MOMENTS

## ABSTRACT

We propose *NovoGrad*, an adaptive stochastic gradient descent method with layer-wise gradient normalization and decoupled weight decay. In our experiments on neural networks for image classification, speech recognition, machine translation, and language modeling, it performs on par or better than well-tuned SGD with momentum, Adam, and AdamW. Additionally, NovoGrad (1) is robust to the choice of learning rate and weight initialization, (2) works well in a large batch setting, and (3) has half the memory footprint of Adam.

## 1 INTRODUCTION

The most popular algorithms for training Neural Networks (NNs) are Stochastic Gradient Descent (SGD) with momentum (Polyak, 1964; Sutskever et al., 2013) and Adam (Kingma & Ba, 2015). SGD with momentum is the preferred algorithm for computer vision, while Adam is the most commonly used for natural language processing (NLP) and speech problems. Compared to SGD, Adam is perceived as safer and more robust to weight initialization and learning rate. However, Adam has certain drawbacks. First, as noted in the original paper (Kingma & Ba, 2015), the second moment can vanish or explode, especially during the initial phase of training. To alleviate this problem, a learning rate (LR) warmup (Goyal et al., 2017) is typically used. Adam often leads to solutions that generalize worse than SGD (Wilson et al., 2017), and to improve Adam regularization, Loshchilov & Hutter (2019) proposed AdamW with decoupled weight decay.

Our motivation for this work was to find an algorithm which: (1) performs equally well for image classification, speech recognition, machine translation, and language modeling, (2) is robust to learning rate and weight initialization, (3) has strong regularization properties. We start with Adam, and then (1) replace the element-wise second moment with the layer-wise moment, (2) compute the first moment using gradients normalized by layer-wise second moment, (3) and decouple weight decay (similar to AdamW) from normalized gradients. The resulting algorithm, *NovoGrad*, combines SGD's and Adam's strengths. We applied NovoGrad to a variety of large scale problems — image classification, neural machine translation, language modeling, and speech recognition — and found that in all cases, it performs as well or better than Adam/AdamW and SGD with momentum.

## 2 RELATED WORK

NovoGrad belongs to the family of **Stochastic Normalized Gradient Descent (SNGD)** optimizers (Hazan et al., 2015; Nesterov, 1984). SNGD uses only the direction of the stochastic gradient to update the weights, and the step size does not depend on the magnitude of that gradient. Hazan et al. (2015) proved that the direction of the gradient was sufficient for convergence. Ignoring the gradient magnitude makes SNGD robust to vanishing and exploding gradients.

SNGD with **layer-wise gradient normalization** was introduced by Singh et al. (2015). The method scales up small gradients, while keeping large gradients unchanged:

$$\hat{\boldsymbol{g}}_t^l = \boldsymbol{g}_t^l \cdot (1 + \log(1 + \frac{1}{||\boldsymbol{g}_t^l||}))$$

where $g_t^l$ is the gradient for the layer $l$ at step $t$. A similar approach was proposed by Yu et al. (2018), who used layer-wise gradient normalization for SGD and Adam to alleviate vanishing and exploding gradients. They divide the stochastic gradient $g_t^l$ by its norm $||g_t^l||$:

$$\hat{g}_t^l = \frac{g_t^l}{||g_t^l||}$$

NovoGrad is also closely related to the **Normalized Direction-preserving Adam (ND-Adam)**, proposed by Zhang et al. (2017). For each layer, ND-Adam first removes the projection of gradient $g_t^l$ on the layer's weights $w_t^l$:

$$\bar{g}_t^l = g_t^l - (g_t^l, w_t^l) \cdot w_t^l$$

Then, $\bar{g}_t^l$ is used to compute Adam 1$^{\text{st}}$ and 2$^{\text{nd}}$ (scalar) moments:

$$m_t^l = \beta_1 \cdot m_{t-1}^l + (1 - \beta_1) \cdot \bar{g}_t^l$$
$$v_t^l = \beta_2 \cdot v_{t-1}^l + (1 - \beta_2) \cdot ||\bar{g}_t^l||^2$$

Similar to Adam, the weights are updated with the 1$^{\text{st}}$ moment re-scaled by the 2$^{\text{nd}}$ moment:

$$\bar{w}_{t+1}^l = w_t^l - \lambda_t \cdot \frac{m_t^l}{\sqrt{v_t^l} + \epsilon}$$

Adaptive **methods like Adam generalize worse than SGD with momentum** as was shown in Wilson et al. (2017). For example, Keskar & Socher (2017) proposed to use Adam during the initial stage only and then switch to SGD. Luo et al. (2019) suggested to improve Adam regularization by limiting the factor $\frac{1}{\sqrt{v_t}}$ to a certain range: limiting from above helps to decrease the training loss while limiting from below helps to generalize better. Loshchilov & Hutter (2019) showed that Adam's weak regularization is due to the fact that the 2$^{\text{nd}}$ moment normalization effectively turns off L2-regularization. They proposed **AdamW**, which **decouples the weight decay** $d \cdot w_t$ from the gradient and uses it directly in the weight update:

$$w_{t+1} = w_t - \lambda_t \cdot \left(\frac{m_t}{\sqrt{v_t} + \epsilon} + d \cdot w_t\right)$$

Adam needs to store the 2$^{\text{nd}}$ moment, and this doubles the optimizer memory compared to SGD with momentum. This affects large models like GPT-2 (Radford et al., 2019) with 1.5 billion parameters. Shazeer & Stern (2018) proposed the **AdaFactor** algorithm, which replaced the full 2$^{\text{nd}}$ moment with moving averages of the row and column sums of the squared gradients. For a layer defined by an $n \times m$ matrix, this would reduce memory from $\mathcal{O}(n \times m)$ to $\mathcal{O}(n + m)$. NovoGrad consumes the same amount of memory as SGD with momentum.

## 3 ALGORITHM

NovoGrad is based on 3 ideas: (1) layer-wise 2$^{\text{nd}}$ moments instead of 2$^{\text{nd}}$ moment per each parameter, (2) gradients normalization with layer-wise 2$^{\text{nd}}$ moments, (3) decoupled weight decay.

Let $g_t^l$ be the stochastic gradient for layer $l$ at step $t$. NovoGrad first computes the layer-wise 2$^{\text{nd}}$ moment $v_t^l$ using the norm $||g_t^l||$:[1]

$$v_t^l = \beta_2 \cdot v_{t-1}^l + (1 - \beta_2) \cdot ||g_t^l||^2 \tag{1}$$

where $0 \le \beta_2 \le 1$. We use much smaller $\beta_2$ than in Adam, usually in the range $[0.2, 0.5]$.[2]

The moment $v_t^l$ is used to normalize the gradient $g_t^l$ before calculating the first moment $m_t^l$. Similarly to AdamW, we decouple weight decay $d \cdot w_t$ from the stochastic gradient, but we add it to normalized gradient before computing moment $m_t^l$:

$$m_t^l = \beta_1 \cdot m_{t-1}^l + \left(\frac{g_t^l}{\sqrt{v_t^l} + \epsilon} + d \cdot w_t\right) \tag{2}$$

---

[1] We use $L_2$-norm $||g_t^l||$ to compute $v_t^l$. It would be interesting to see how $L_1$ or $L_\infty$ norms perform.
[2] If $\beta_2 = 0$, then $v_t^l = ||g_t^l||^2$, and NovoGrad becomes layer-wise NGD with decoupled weight decay.

---

**Algorithm 1** NovoGrad

> **Parameters:** Initial learning rate $\lambda_0$, moments $\beta_1, \beta_2$, weight decay $d$, number of steps $T$
> **Weight initialization:** $t = 0$, Initialize $\boldsymbol{w}_0$.
>
> **Moment initialization:** $t = 1$, for each layer $l$ set $v_1^l = ||\boldsymbol{g}_1^l||^2; \boldsymbol{m}_1^l = \frac{\boldsymbol{g}_1^l}{\sqrt{v_1^l}} + d \cdot \boldsymbol{w}_0^l$.
>
> **while** $t \leq T$ **do**
>    $\lambda_t \leftarrow LearningRateUpdate(\lambda_0, t, T)$ (compute the global learning rate)
>    **for** each layer $l$ **do**
>       $\boldsymbol{g}_t^l \leftarrow \nabla_l L(w_t)$
>       $v_t^l \leftarrow \beta_2 \cdot v_{t-1}^l + (1 - \beta_2) \cdot ||\boldsymbol{g}_t^l||^2$
>       $\boldsymbol{m}_t^l \leftarrow \beta_1 \cdot \boldsymbol{m}_{t-1}^l + (\frac{\boldsymbol{g}_t^l}{\sqrt{v_t^l + \epsilon}} + d \cdot \boldsymbol{w}_t^l)$
>       $\boldsymbol{w}_{t+1}^l \leftarrow \boldsymbol{w}_t^l - \lambda_t \cdot \boldsymbol{m}_t^l$
>    **end for**
> **end while**

---

where $0 < \beta_1 < 1$ is the momentum, typically in the same range as in SGD or Adam $[0.9 - 0.95]$. The first moment can be also computed via an exponential moving average in Adam-like style:

$$\boldsymbol{m}_t^l = \beta_1 \cdot \boldsymbol{m}_{t-1}^l + (1 - \beta_1) \cdot (\frac{\boldsymbol{g}_t^l}{\sqrt{v_t^l + \epsilon}} + d \cdot \boldsymbol{w}_t^l)$$

Finally, weights are updated the same way as in SGD with momentum.

Similar to Adam, one can construct a counter-example for NovoGrad in the stochastic convex optimization settings (Wilson et al., 2017). However, the "AMS-Grad" fix (Reddi et al., 2018) for Adam can also be applied in this case to guarantee NovoGrad convergence:

$$v_t^l = \beta_2 \cdot v_{t-1}^l + (1 - \beta_2) \cdot ||\boldsymbol{g}_t^l||^2$$
$$\hat{v}_t^l = \max(\hat{v}_{t-1}^l, v_t^l)$$
$$\boldsymbol{m}_t^l = \beta_1 \cdot \boldsymbol{m}_{t-1} + (\frac{\boldsymbol{g}_t^l}{\sqrt{\hat{v}_t^l + \epsilon}} + d \cdot \boldsymbol{w}_t^l)$$

## 4 EXPERIMENTS WITH DEEP LINEAR NETWORKS

Following (Andrew M. Saxe & Ganguli, 2013; Ian J. Goodfellow & Saxe, 2015) we will use Novo-Grad to train linear model composed of two linear layers $w_1, w_2$ without any non-linearity. The model $y = (w_1 \cdot w_2) \cdot x$ should output 1 when $x = 1$. This model is linear with respect to the inputs, but it is non-linear with respect to the weights, since they are factorized into the product of layers' weights. Training the model is equivalent to the minimization of the loss $L(w_1, w_2) = (w_1 \cdot w_2 - 1)^2$ (Ian J. Goodfellow & Saxe, 2015). The loss is not convex, and its minima are located on the hyperbola $w_1 w_2 = 1$ (see Figure 1). Minima close to the points $(-1, -1)$ and $(1, 1)$ are good "flat" minima which generalize well. Minima close to the axes are "sharp" minima (Keskar et al., 2016).

We trained the model with SGD with momentum, Adam, AdamW, and NovoGrad, using the same fixed learning rate,[3] weight decay, and weights initialization. The model was trained for 500 steps. Figure 2 shows the training trajectory and the zoomed-out area near the final point. All algorithms behave in a similar way: first the trajectory goes to the curve $w_2 = 1/w_1$, and then follows the hyperbola towards $(1, 1)$ or $(-1, -1)$. During the first phase, training loss decreases, and during the second phase, generalization improves. SGD converges nicely toward $(1, 1)$ but its trajectory is still slightly off of the optimal solution. Adam oscillates wildly around hyperbola $w_2 = 1/w_1$, while AdamW behaves much better since weight decay decoupling significantly reduces oscillations.

NovoGrad is the most stable out of four algorithms. It exhibits better generalization and closely follows the minima curve because normalized gradients prevent trajectory from going far from it. We also found that NovoGrad is more robust than other algorithms to the choice of learning rate, weight decay, and weight initialization (see for details Appendix A).

---

[3]We will use the gradient averaging for SGD first moment as in Adam: $m_t = \beta \cdot m_{t-1} + (1 - \beta) \cdot g_t$ to use the same LR for all optimizers.

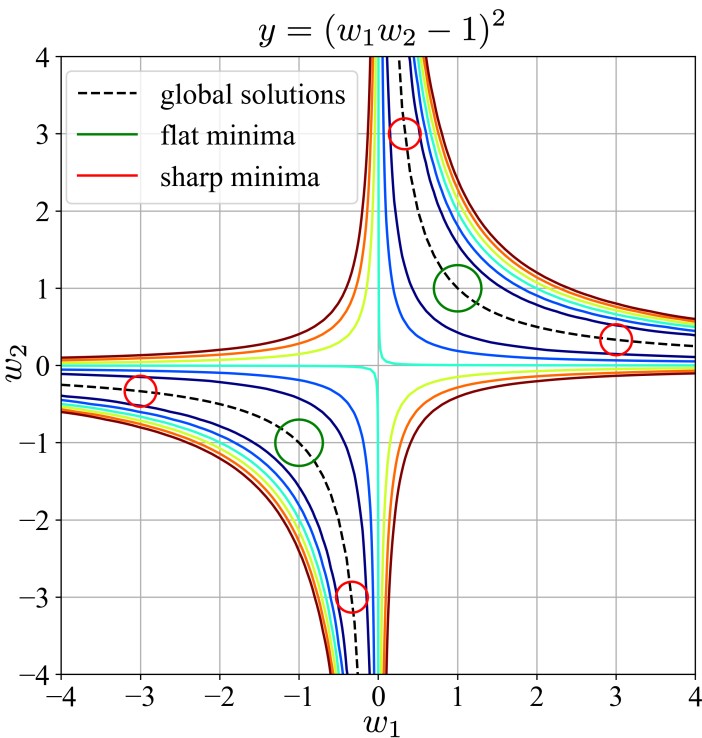

Figure 1: 2D-contour plot of surface $y = (w_1 \cdot w_2 - 1)^2$ of linear model with two layers. The loss functions has many global minima are located on hyperbola $w_2 = 1/w_1$. Solutions near $(-1, 1)$ and $(1, 1)$ are good "flat" minima, and solutions near axes are "sharp" minima.

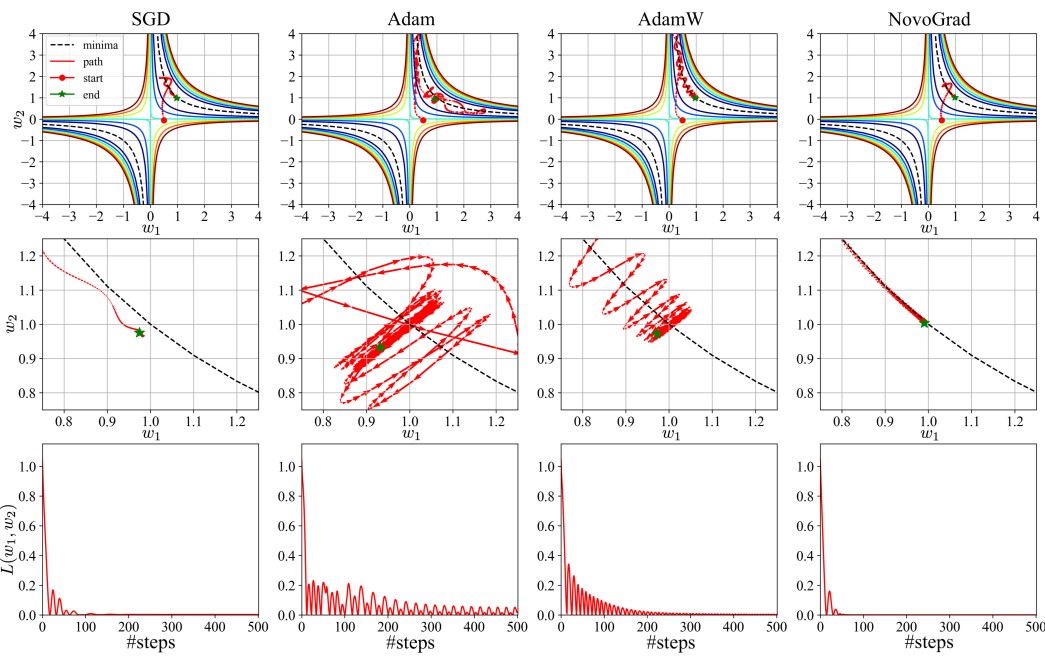

Figure 2: Deep Linear Network with two layers: training with SGD, Adam, and NovoGrad

# 5 EXPERIMENTS WITH LARGE DNNS

We train four deep models: ResNet-50 (He et al., 2016) — for ImageNet classification, Transformer-big (Vaswani et al., 2017) — for WMT 2014 translation, Jasper (Li et al., 2019) — for LibriSpeech speech recognition, and Transformer-XL (Dai et al., 2019) — for WikiText-103 word-level language modeling, with NovoGrad, SGD with momentum, and Adam/AdamW.[4] Each model was trained on a single DGX-1 machine with 8 NVIDIA V100 GPUs with gradient accumulation used for large batch training. In all the experiments, NovoGrad performed on par or better than other algorithms.

## 5.1 IMAGE CLASSIFICATION

We used ResNet-50 v2 (He et al., 2016) for ImageNet classification task (Russakovsky et al., 2015). We trained this model with three optimizers: SGD with momentum (SGD), AdamW, and NovoGrad. All models have been trained with the batch size of 1024 for 100 epochs. We used quadratic LR decay for SGD with momentum and cosine decay (Loshchilov & Hutter, 2016) for AdamW and NovoGrad. We could not find any training recipe for ResNet-50 with AdamW, so we report the best accuracy we achieved after extensive hyper-parameter search. We used only standard data augmentation methods: re-size, flip, random crop, and did not employ any additional training tricks (He et al., 2018). The single-crop validation accuracy for each algorithm is reported in Table 1.

Table 1: ImageNet classification — ResNet-50 v2, batch 1024, top-1 and top-5 accuracy(%).

| Optimizer | LR policy | Init LR | Weight decay | Epochs | top-1,% | top-5,% |
|:---:|:---:|:---:|:---:|:---:|:---:|:---:|
| SGD | poly (2) | 0.400 | 0.0001 | 100 | 76.38 | 93.08 |
|  |  |  |  | 200 | 76.33 | 92.96 |
| AdamW | cosine | 0.002 | 0.120 | 100 | 76.36 | 93.01 |
|  |  |  |  | 200 | 76.48 | 92.94 |
| NovoGrad | cosine | 0.007 | 0.002 | 100 | 76.94 | 93.41 |
|  |  |  |  | 200 | 77.74 | 93.70 |
|  |  |  |  | 300 | 77.65 | 93.62 |

NovoGrad outperformed both AdamW and SGD with the top-1 accuracy of 76.94% after 100 epochs. SGD and Adam accuracy remained under 76.5% if we trained for 200 epochs instead, while NovoGrad accuracy improved to 77.74%. NovoGrad demonstrated powerful regularization capabilities: training for 100 additional epochs kept top-1=77.65% without overfitting. Note that this is "vanilla" ResNet-50, without sophisticated data augmentation or model tweaking.

### 5.1.1 LARGE BATCH TRAINING FOR IMAGE CLASSIFICATION

Hazan et al. (2015) showed that large batch size is beneficial for SNGD convergence, which motivated us to explore NovoGrad for large batch training. We trained ResNet-50 v2 with batch sizes of 8K and 32K. To compare with the previous methods, we train the model for 90 epochs using cosine LR decay. To emulate a large batch, we used a mini-batch of 128 per GPU and accumulated gradients from several mini-batches before each weight update.

Table 2: Large batch training with NovoGrad — ImageNet, ResNet-50 v2, 90 epochs, accuracy(%).

| Batch | Init LR | Weight decay | Top-1,% | Top-5,% |
|:---:|:---:|:---:|:---:|:---:|
| 1K | 0.070 | 0.002 | 76.86 | 93.31 |
| 8K | 0.016 | 0.006 | 76.64 | 93.14 |
| 32K | 0.026 | 0.010 | 75.78 | 92.54 |

To establish the baseline for NovoGrad training with batch 32K we first used the method similar to proposed in Goyal et al. (2017): scaling the learning rate linearly with the batch size and using

---

[4]Training was done in OpenSeq2Seq (Kuchaiev et al., 2018): `https://github.com/NVIDIA/OpenSeq2Seq/blob/master/example_configs/image2label/resnet-50-v2-mp.py`.

LR warmup. This method gives top-1=75.09% and top-5=92.27%. We found that we get much better results when we increase both the learning rate $\lambda$ and the weight decay $d$ to improve the regularization (see Table 2).

For comparison, we took 3 methods, which (1) use fixed batch size during training and (2) do not modify the original model. All 3 methods employ SGD with momentum. The first method (Goyal et al. (2017)) scales LR linearly with batch size and uses the LR warmup to stabilize the initial training phase. The second method (You et al. (2018)) combines warmup with Layer-wise Adaptive Rate Scaling (LARS) (You et al., 2017). The last method (Codreanu et al. (2017)) uses warmup and dynamic weight decay (WD). NovoGrad outperformed all other methods without using any additional techniques like LR warmup (Goyal et al., 2017), dynamic weight decay, special batch normalization initialization, etc. Using warm-up (500 steps) we slightly improved top1 accuracy to 75.99% and top5 to 92.72%.

Table 3: Large batch training comparison — ImageNet, ResNet-50v 2, top-1 accuracy(%) .

| Optimizer | Reference | Bag of Tricks | Epochs | B=8K | B=32K |
|---|---|---|---|---|---|
| SGD | Goyal et al. (2017) | warmup | 90 | 76.26 | 72.45 |
| SGD | You et al. (2018) | warmup, LARS | 90 | 75.30 | 75.40 |
| SGD | Codreanu et al. (2017) | warmup, multi-step WD | 100 | 76.60 | 75.31 |
| NovoGrad | | — | 90 | 76.64 | 75.78 |
| | | warmup | 90 | — | 75.99 |

## 5.2 SPEECH RECOGNITION

We conducted experiments with Jasper-10x5 (Li et al. (2019)), a very deep convolutional neural acoustic model, on the LibriSpeech speech recognition task (Panayotov et al., 2015). Jasper was trained with SGD with momentum (SGD), Adam and NovoGrad for 400 epochs with a batch of 256, polynomial LR decay, and Layerwise Adaptive Rate Clipping (LARC).[5] We found that NovoGrad yields lower Word Error Rates (WER) comparing to SGD, especially for the long runs. The model and training parameters are described in Li et al. (2019).

Table 4: Speech recognition — Jasper-10x5, LibriSpeech, 400 epochs, WER (%).

| Optimizer | Dev | | Test | |
|---|---|---|---|---|
| | clean | other | clean | other |
| Adam | 13.20 | 31.71 | 13.36 | 32.71 |
| SGD | 3.91 | 12.77 | 3.98 | 12.79 |
| NovoGrad | 3.64 | 11.89 | 3.86 | 11.95 |

### 5.2.1 LARGE BATCH TRAINING FOR ASR

We trained Jasper10x5 with batch sizes of 512, 4K, 8K, 16K and 32K on LibriSpeech. In all cases, we trained the model for 400 epochs. For batch size up to 8K we scaled LR linearly with the batch size and used LR warmup. To scale batch to 16K and 32K we also increased weight decay (see Table 5). The batch 16K leads to WER comparable to the baseline. Batch 32K has higher WER due to the smaller number of training steps (9 weights updates per epoch). Figure 3 shows WER on dev-clean during training for different batch sizes.

## 5.3 LANGUAGE MODELING

We trained Transformer-XL (Dai et al., 2019), the state-of-the-art LM architecture on the word-level WikiText–103 (Merity et al., 2016) benchmark. For all the experiments we used a 16-layer

---

[5]https://github.com/NVIDIA/apex/blob/master/apex/parallel/LARC.py.

Table 5: Large batch training with NovoGrad — Jasper-10x5, LibriSpeech, 400 epochs, WER (%).

| Batch | Init LR | Warmup | Weight decay | Dev clean | Dev other | Test clean | Test other |
|-------|---------|--------|--------------|-----------|-----------|------------|------------|
| 512 | 0.015 | - | 0.001 | 3.58 | 11.30 | 3.85 | 11.29 |
| 4K | 0.03 | 0.05 | 0.001 | 3.66 | 11.67 | 3.92 | 11.68 |
| 8K | 0.06 | 0.05 | 0.001 | 3.69 | 11.76 | 3.96 | 11.75 |
| 16K | 0.06 | 0.05 | 0.003 | 3.67 | 11.03 | 3.94 | 11.19 |
| 32K | 0.06 | 0.08 | 0.004 | 4.01 | 11.73 | 4.14 | 11.89 |

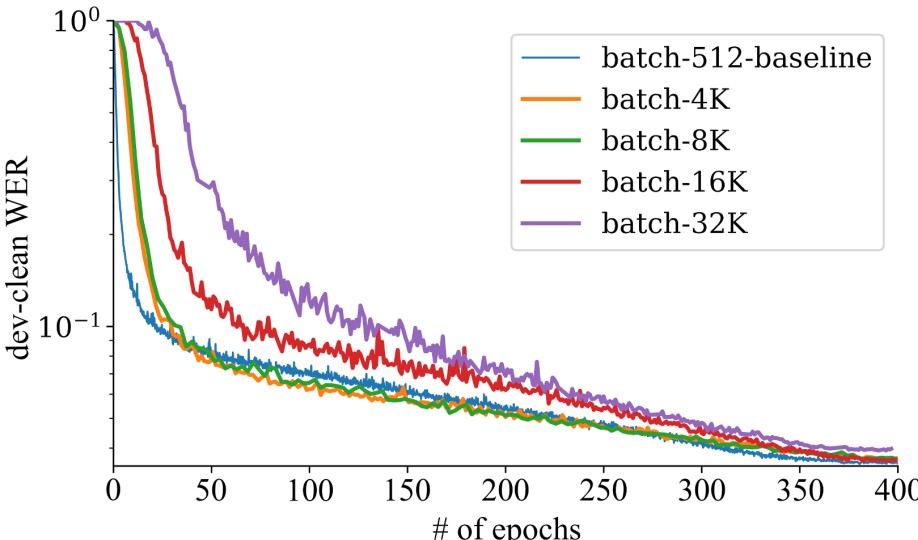

Figure 3: ASR, large batch training. Jasper-10x5 trained with NovoGrad on LibriSpeech.

model with 191M parameters ($d_{model} = 512$, $d_{ff} = 2048$, $h = 8$, $P_{drop} = 0.15$). All other hyperparameters were taken from the original Transformer-XL paper, the source code was based on a publicly available implementation.[6] Each configuration was trained for 12 billion tokens which is approximately 117 epochs and 366K training iterations. Figure 4 shows that NovoGrad exhibits a much smaller gap between training and validation perplexity compared to Adam, which results in better performance on the test set. Longer training for 20B tokens does not lead to overfitting as the resulting validation and test perplexities improve even further.

Table 6: LM. Transformer-XL trained on WikiText-103 with batch size 256, sequence length 512.

| Optimizer | Tokens | Init LR | Weight decay | Val PPL | Test PPL |
|-----------|--------|---------|--------------|---------|----------|
| Adam | 12B | 0.00025 | - | 23.84 | 25.40 |
| AdamW | 12B | 0.00025 | 0.001 | 23.64 | 25.06 |
| NovoGrad | 12B | 0.01 | 0 | 20.53 | 21.26 |
| | 20B | 0.01 | 0 | 19.89 | 20.65 |

---

[6]https://github.com/cybertronai/transformer-xl

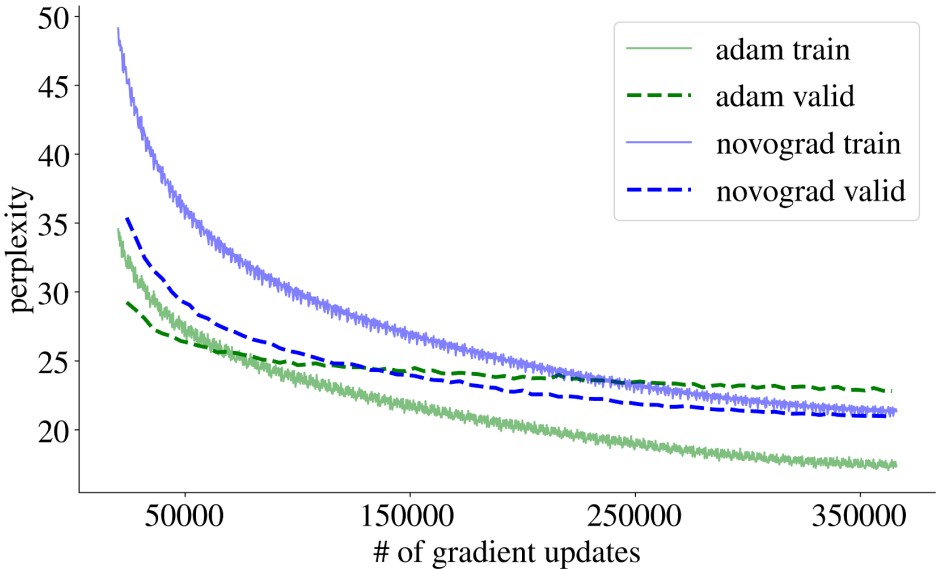

Figure 4: LM. Transformer-XL trained with Adam and NovoGrad on WikiText-103.

## 5.4 NEURAL MACHINE TRANSLATION

We trained Transformer (Vaswani et al., 2017) on WMT 2014 English-to-German benchmark. For all the experiments, we used a 12-layer Transformer-big model with 185M parameters ($d_{model} = 1024$, $d_{ff} = 4096$, $h = 16$) with the vocabulary of 8192 tokens based on joint source-target byte-pair-encodings (Sennrich et al., 2015). For Adam and AdamW we used dropout of $P_{drop} = 0.3$ and for NovoGrad we used $P_{drop} = 0.2$. We trained all algorithms with mixed-precision (Micikevicius et al., 2017) for 100K steps (approximately 150 epochs) with a 4K steps warmup on batches of up to 490K source and target tokens obtained via gradient accummulation (Ott et al., 2018) with cosine learning rate annealing. We did not use checkpoint averaging, all the results are reported for the last checkpoint in the corresponding run.

Table 7: WMT'14 English-to-German translation, Transformer-big, batch 490K tokens, 150 epochs, no checkpoint averaging. Tokenized BLEU and detokenized SacreBLEU on WMT'14 (newstest14).

| Optimizer | Init LR | Weight decay | SacreBLEU | TokenBLEU |
|-----------|---------|--------------|-----------|-----------|
| Adam | 0.0006 | - | 28.26 | 28.71 |
| AdamW | 0.0006 | 0.005 | 28.24 | 28.72 |
| NovoGrad | 0.04 | 0.0001 | 28.80 | 29.35 |

## 6 CONCLUSION

We propose NovoGrad – an adaptive SGD method with gradients normalized by the layer-wise second moment and with decoupled weight decay. We tested NovoGrad on deep models for image classification, speech recognition, translation, and language modeling. In all experiments, NovoGrad performed equally or better than SGD and Adam/AdamW. NovoGrad is more robust to the initial learning rate and weight initialization than other methods. For example, NovoGrad works well without LR warm-up, while other methods require it. NovoGrad performs exceptionally well for large batch training, e.g. it outperforms other methods for ResNet-50 for all batches up to 32K. In addition, NovoGrad requires half the memory compared to Adam.

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

## A  TRAINING OF DEEP LINEAR NETWORKS

Following (Andrew M. Saxe & Ganguli, 2013; Ian J. Goodfellow & Saxe, 2015) we will use Novo-Grad to train linear NN model composed of two linear layers $w_1, w_2$ without any non-linearity. The model $y = (w_1 \cdot w_2) \cdot x$ should output 1 when $x = 1$. The model is linear function of the inputs, but it is non-linear function of the weights, since they are factorized into the product of layers' weights. Training the model is equivalent to the minimization of the loss $L(w_1, w_2) = (w_1 \cdot w_2 - 1)^2$. The loss is not convex, and its minima are located on the curve: $w_1 w_2 = 1$. Minima close to the points $(-1, -1)$ and $(1, 1)$ are good "flat" minima which generalize well. Minima close to axes are "sharp" minima with bad generalization (see (Keskar et al., 2016)). The 2D-contour plot of the loss function shown on Figure 5.

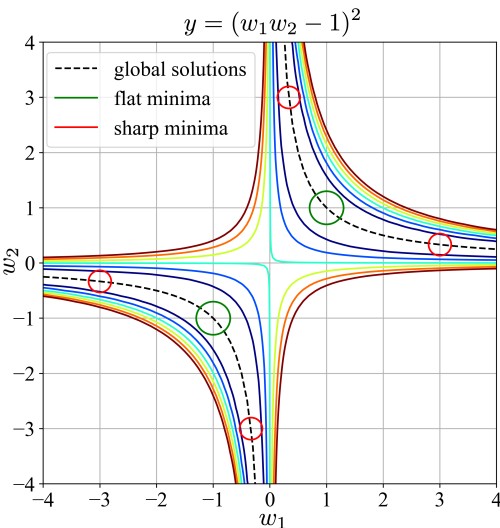

Figure 5: 2D-contour plot of surface $y = (w_1 \cdot w_2 - 1)^2$ of linear model with two layers. The loss functions has many global minima located on hyperbola $w_2 = 1/w_1$. Solutions near $(-1, -1)$ and $(1, 1)$ are good "flat" minima, and solutions near axes are "sharp" minima.

We will study how the behavior of each algorithm depends on learning rate, weight decay and initialization. We will train the model with each optimizer for 500 steps using the same learning rate, weight decay, and weights initialization. To use the same learning rate for all optimizers, we will use the "gradient averaging" for NovoGrad. We will also use the version of SGD with "gradient averaging" (similar to Adam): $m_t = \beta \cdot m_{t-1} + (1 - \beta) \cdot g_t$. For fixed learning rate this SGD version is equivalent to the regular SGD with momentum.

Training trajectories for the baseline (fixed learning rate 0.2, weight decay 0.1, and $\beta_1 = 0.95, \beta_2 = 0.5$.) are shown on the Figure 6. All algorithms first go to the curve $w_2 = 1/w_1$, and then slide along hyperbola towards $(1, 1)$ or $(-1, -1)$. SGD is slightly off with respect to the optimal solution. Adam oscillates wildly around line $w_2 = 1/w_1$. AdamW behaves better since weight decay decoupling significantly reduces osculations. NovoGrad is the most stable out of four algorithms, it also shows much better generalization than other algorithms and converges to $(1, 1)$ closely following the minima curve.

Next, we increased learning rate from 0.2 to 1.0 while keeping weight decay equal to 0.1. Training trajectories are shown on the Figure 7: SGD and Adam diverge. Only AdamW and NovoGrad converge.

Similarly, when we increased weight decay from 0.1 to 0.5 while keeping learning rate 0.2, all algorithms except NovoGrad diverge, while NovoGrad demonstrates high robustness to the weight decay choice (see Figure 8).

Finally, we started training from different initial point. SGD and NovoGrad are most robust with respect to the initialization, while AdamW diverge (see Figure 9).

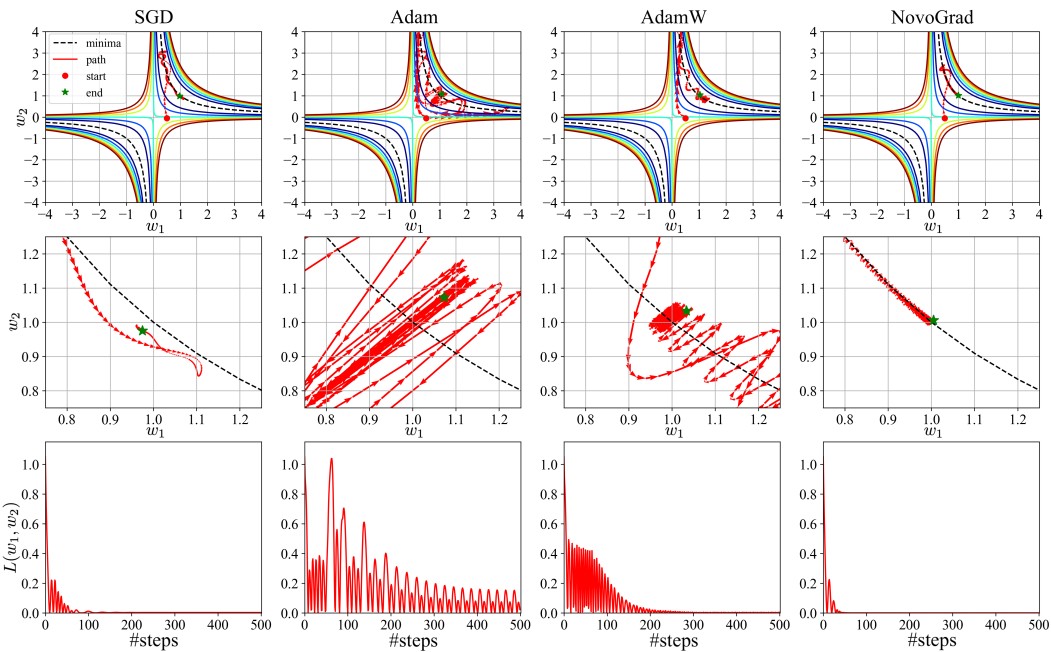

Figure 6: DLN training – baseline: learning rate 0.2, weight decay 0.1.

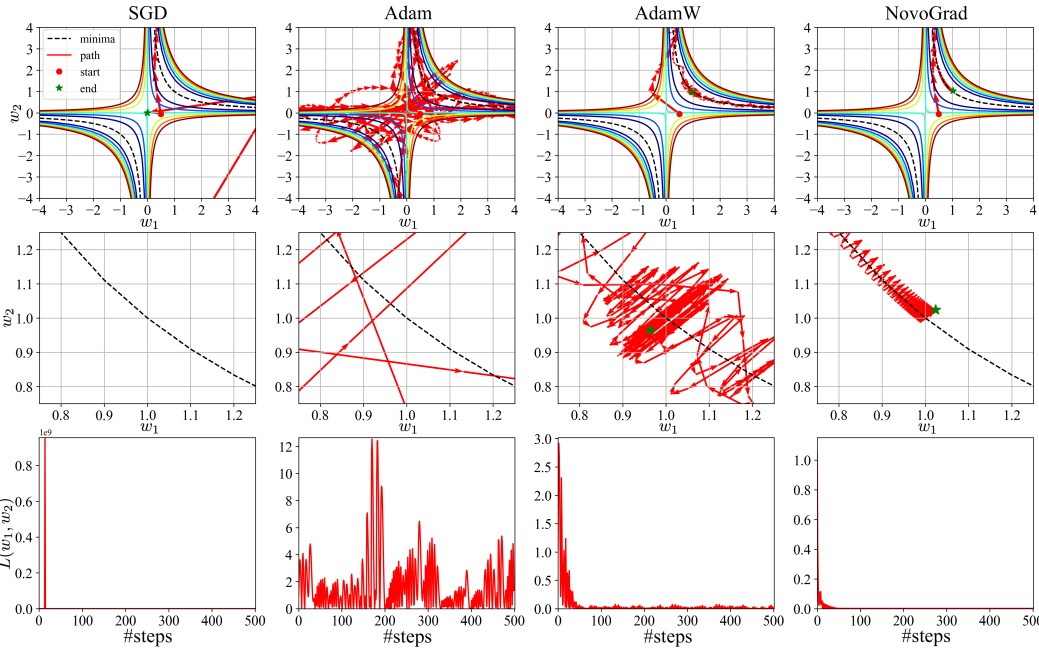

Figure 7: DLN training – increased learning rate $0.2 \rightarrow 1.0$: SGD and Adam diverges, AdamW and NovoGrad converge.

To summarize our experiments with linear neural network: NovoGrad is more robust than other algorithms to the choice of learning rate, weight decay, and weight initialization.

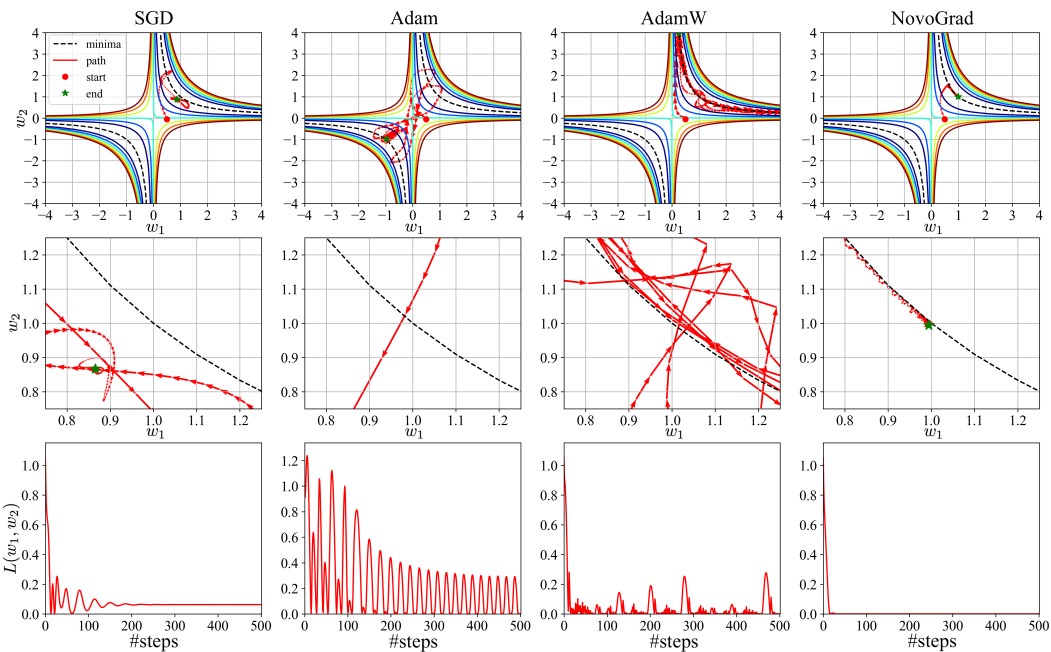

Figure 8: DLN training – increased weight decay $0.1 \rightarrow 0.5$: All algorithms except NovoGrad diverge.

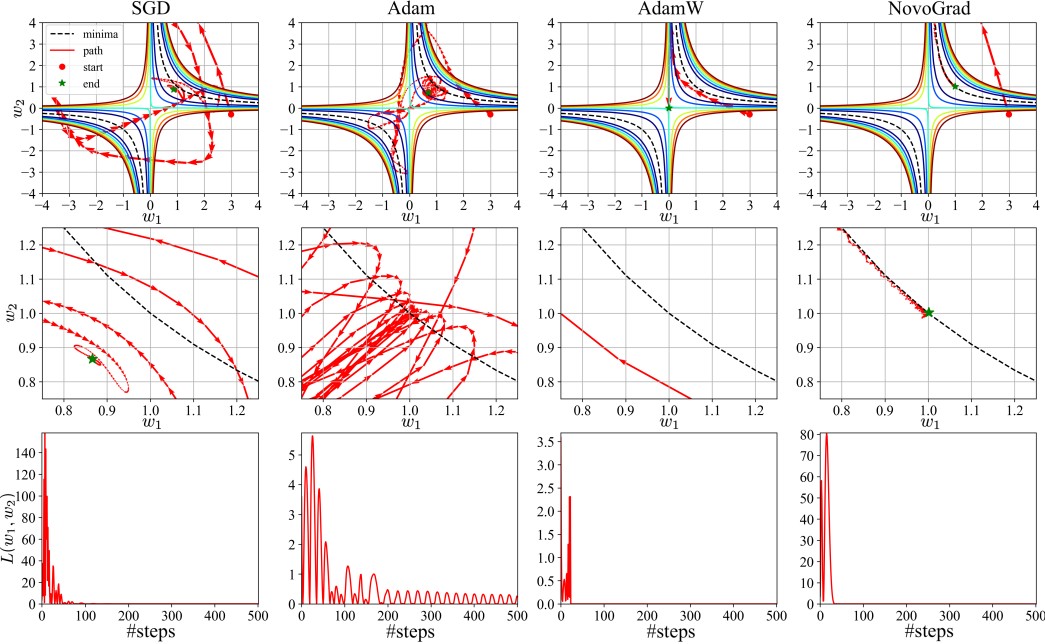

Figure 9: DLN training – "bad" initialization: AdamW diverges.

