# OpenReview forum: "Training Deep Networks with Stochastic Gradient Normalized by Layerwise Adaptive Second Moments"
_ICLR.cc/2020/Conference — Reject_

### Official Review · AnonReviewer3 · 2019-10-22
**Official Blind Review #3**

**Rating:** 6

**Review:**

The authors present a variation of Adam that combines layerwise normalization and weight decay decoupling. They test quite extensively their algorithm against other optimization methods on varied tasks.

From a theoretical point of view, the contribution seems very incremental. The authors acknowledge that weight decay decoupling is already present in the literature, so it appears that the only contribution is the layer-wise normalization. No justification is proposed as to why this kind of normalization would either accelerate the convergence or leads to better generalization. Proof of convergence even in a deterministic convex setting are missing, so the reader has to extrapolate correctness from previous work on adaptive gradient descent.
On the other hand, the proposed algorithm is tested on a great variety of tasks, using state of the art models. The reported performance for the benchmarks (checked for Resnet and Transformer-XL) are on par with what can be found in the literature. The proposed method outperform consistently the other optimizers in terms of generalization performance.

I have some concerns regarding section 4:
 “SGD converges nicely toward (1, 1) but its trajectory is still slightly off of the optimal solution”. It is unclear to me what the reader should understand. Does it converge to the optimal solution? If yes, why should we expect the trajectory to follow the hyperbola?
Using the same learning rate for all methods is a bit odd. Why not search for the best learning rate for each optimizer and report its performance? It seems that the oscillation of some of the optimizers could be fixed by using a smaller learning rate. Also, it can be seen in section 5 that Adam consistently needs a smaller learning rate than Novograd.
If the difference between NovoGrad and AdamW lies in the layer-wise second moment, it is unclear to me why their performance should differ on this task, as each layer has only one weight. It would be great if the authors could clarify this point.

As a conclusion, I am a bit conflicted regarding this paper. The motivation for this modified version of AdamW are unclear, but the empirical results are convincing and rigorous. The authors made a great effort in testing in a variety of different settings. I’m leaning toward accepting this paper, to give the community a chance of testing and, maybe, adopting it.


**Experience Assessment:**

I have read many papers in this area.

**Review Assessment: Checking Correctness Of Derivations And Theory:**

N/A

**Review Assessment: Checking Correctness Of Experiments:**

I assessed the sensibility of the experiments.

**Review Assessment: Thoroughness In Paper Reading:**

I read the paper at least twice and used my best judgement in assessing the paper.

---

> ### Author Response · Authors · 2019-11-15
> **Response to reviewer 3**
>
> Thanks for detailed feedback!
>
> Q1: "No justification is proposed as to why this kind of normalization would either accelerate the convergence or leads to better generalization."
> A1: The gradient normalization accelerate the convergence since it makes algorithm robust wrt very large gradients and wrt vanishing gradients (e.g. when the trajectory is close to  saddle points).  The improved generalization requires both decoupled weight decay and gradient normalization, which keep trajectory close to the minima manifold.
>
> Q2: “SGD converges nicely toward $(1, 1)$ but its trajectory is still slightly off of the optimal solution”. It is unclear to me what the reader should understand. Does it converge to the optimal solution? If yes, why should we expect the trajectory to follow the hyperbola?"
> A2: We used the toy linear NN problem to illustrate that the training consists of two phases for all 4 algorithms -- SGD, Adam, AdamW, NovoGrad:
> 1. phase 1 "Training loss minimization":  the trajectory goes from the initial point to some solution on the minima manifold, given by equation $w1*w2=1$
> 2. phase 2 "Regularization":  trajectory goes from that solution to the good solution along the minima manifold.
> Points $(1,1)$ and $(-1,-1)$ are better from the regularization point of view since Hessian at any minima has 2 eigen values: $0$ and $(w_1^2 + w_2^2)$. We want the solution with minimal largest eigen value, and there are two such minima: $(-1,-1)$ and $(1,1)$.
> Note that the second phase "regularization" requires explicit weight decay / L2 regularization. For example, if we remove weight decay the trajectory stays at the point where it meets the minima manifold. This is true for all algorithms (SGD , NovoGrad...). The trajectory follows the hyperbola (minima manifold) since weight decay pulls is toward the origin, but loss gradient doesn't let trajectory to go too far from the minima manifold. Note that for NovoGrad the penalty for leaving the minima manifold is much higher than for SGD, and the trajectory stays more close to the hyperbola.
>
> Q2: "Using the same learning rate for all methods is a bit odd. Why not search for the best learning rate for each optimizer and report its performance? It seems that the oscillation of some of the optimizers could be fixed by using a smaller learning rate."
> A2: We used the same LR for all optimizers just to illustrate qualitative difference in their behavior.
>
> Q3:"it can be seen in section 5 that Adam consistently needs a smaller learning rate than NovoGrad."
> A3: Correct, we found that Adam performs best with learning rate smaller than learning rate used for NovoGrad. See A3 in the response to the 1st reviewer.
>
> Q4: "If the difference between NovoGrad and AdamW lies in the layer-wise second moment, it is unclear to me why their performance should differ on this task, as each layer has only one weight. It would be great if the authors could clarify this point."
> A4: The main difference between AdamW and NovoGrad is that NovoGrad normalize gradients before it compute first moment, while AdamW first computes the first moment, and then normalize it by second moment. This change in order makes NovoGrad more robust to the "gradients outlier", while AdamW keeps remembering very high gradient for long period.

---

### Official Review · AnonReviewer2 · 2019-10-23
**Official Blind Review #2**

**Rating:** 3

**Review:**

In the paper, the authors propose a novel optimization method for training deep learning models. The idea is from the LARS and AdamW. The authors then test the proposed method on multiple experiments, results showing that the proposed method works better than other compared methods.  The following are my concerns:

1) No convergence guarantee in the paper. There are too many papers claiming faster convergence these days, proof of convergence guarantee is always preferred.
2) The proposed method is straightforward and easy to understand. It is just a combination of AdamW and LARS. I am worried about the novelty of this paper.
3) In the experiments, why the compared methods are usually different. For example, compared methods are Adam, SGD, and NovoGrad in table 4 and compared methods are Adam, AdamW, and NovoGrad in table 6.  Why not compare all these methods?
4) When the batch size varies,  is it required to tune beta_2 accordingly? I didn't find it clearly mentioned in the paper, could authors explain how to set it?
5)  I am confusing that NovoGrad method works much better than Adam or AdamW in Table 6 with no weight decay, more explanations are required.
6) It is unclear why NovoGrad is better than LARS. LARS normalizes learning rate through |w|_2/|g|_2. The authors should explain why normalizes using layerwise |g|_2 is better.

Although the idea is straightforward, the proposed method may be helpful for the community.   I will consider increasing the score if authors can address my concerns.

**Experience Assessment:**

I have published one or two papers in this area.

**Review Assessment: Checking Correctness Of Derivations And Theory:**

N/A

**Review Assessment: Checking Correctness Of Experiments:**

I carefully checked the experiments.

**Review Assessment: Thoroughness In Paper Reading:**

I read the paper at least twice and used my best judgement in assessing the paper.

---

> ### Author Response · Authors · 2019-11-15
> **Response to reviewer 2**
>
> Thank you for the review. Clearly we could  do better comparison of NovoGrad with LARS and explain why removed scaling of normalized gradients by $|w|$.
>
> Q1:"There is no convergence guarantee in the paper. There are too many papers claiming faster convergence these days, proof of convergence guarantee is always preferred."
> A1: The proofs of convergence for the majority of optimizers (e.g. Adam, AdaFactor, LARS etc) are given for convex / quasi-convex case only, and usually these proofs closely follow the original proof from Adagrad paper [1].  The proof for Stochastic Normalized GD was also given only in  quasi-convex setting (Hazan et al , 2014 [2]). This proof can be extended in straight-forward  way for NovoGrad. The convexity / quasi-convexity assumptions that don't hold for deep networks.
> The convergence proof for deep networks with more  than 2 layers is open problem. As far as I know, even for vanilla gradient descent  the convergence proof for deep networks was  proposed only for  deep linear networks in Aurora et al , 2018 [3].  We are working to extend their proof for Normalized Gradients type algorithms.
>
> Q2: "The proposed method is just a combination of AdamW and LARS. I am worried about the novelty of this paper."
> A2:   NovoGrad comparison  to  LARS:
> 1. LARS uses norm of layer gradient for normalization. NovoGrad uses the norm of  second moments for the layer. So if we set $\beta_2=0$, the norm of second moment will become just norm of gradient. So NovoGrad is more general comparing to LARS.
> 2. After normalization, LARS rescales the update proportional to the norm of layer weight $|w|_2$. NovoGrad doesn't . The reasons why we removed this rescaling are explained in the A6 below.
> NovoGrad comparison  to  AdamW:
> The main difference between AdamW and NovoGrad is that NovoGrad normalize gradients before it compute first moment, while AdamW first computes the first moment, and then normalize it by second moment. This change in order makes NovoGrad more robust to the "gradients outlier", while AdamW keeps remembering very high gradient for long period.
>
> Q3: "Why the compared methods are usually different. For example, compared methods are Adam, SGD, and NovoGrad in table 4 and compared methods are Adam, AdamW, and NovoGrad in table 6.  Why not compare all these methods?"
> A: The choice of baseline algorithms for each particular problem was based on the best performing optimizers from the literature. We tried to solve several tasks with “non-traditional” optimizers but did not succeed. For example, we could not make Adam converge on ResNet-50 to reasonable accuracy and we could not make SGD converge on Transformer NMT.
>
> Q4: "When the batch size varies,  is it required to tune $\beta_2$ accordingly?"
> A: No. We didn’t use $\beta_2$ tuning for different batch sizes. The default suggested value is $\beta_2=0.25$ which we used in the majority of our experiments (ASR, LM, NMT). ResNet-50 experiments were conducted with  the earlier version of the code with $\beta_2$=0.98
>
> Q5:"Why NovoGrad method works much better than Adam or AdamW in Table 6 with no weight decay?"
> A: For language modeling with Transformer-XL, we used only Dropout for regularization, following the original paper [4]. We experimented with weight decay too, but did not manage to get better results for both NovoGrad and Adam (the scores of AdamW are comparable to those of Adam).
>
> Q6: "Why NovoGrad is better than LARS. LARS normalizes learning rate through $|w|_2/|g|_2$. The authors should explain why normalizes using layer-wise $|g|_2$ is better."
> A: The main weakness of LARS is its behavior in the regions with either too small or too large weights. If weights are near 0, then scaling by $|w|_2$ will make update very small, and it will take too long to leave this region. If weights are large, then the update (which is proportional to the weights norm) can be too large, which might cause instability. NovoGrad does not have this weakness. This is major reason why we removed scaling of normalized gradients by $|w|$ in NovoGrad.
>
> References:
> [1] J. Duchi, E. Hazan, Y. Signer. Adaptive subgradient methods for online learning and stochastic optimization, 2011.
> [2] E. Hazan, K. Levy, S. Shalev-Shwartz. Beyond Convexity: Stochastic Quasi-Convex Optimization, 2014
> [3] S. Arora, N. Cohen, N. Golowich, W.Hu. A Convergence Analysis of Gradient Descent for Deep Linear Neural Networks, 2018.
> [4] Z. Dai, Z. Yang, Y. Yang, J. Carbonell, Q. Le, R. Salakhutdinov. Transformer-XL: language models beyond a fixed-length context, 2019.

---

### Official Review · AnonReviewer1 · 2019-10-24
**Official Blind Review #1**

**Rating:** 3

**Review:**

I am not sure about my assessment of this paper.
The authors propose their approach as a mixture of two other approaches. Then, they proceed with an illustrative 2-D example where they apply the same hyperparameters for all tested methods. This does not seem appropriate especially given that later in the paper they show that SGD with momentum and Adam use very different hyperparameter settings such as learning rate and weight decay. Basically, SGD and NovoGrad perform very similar in Figure 6 and the difference might be due to a too large learning rate for SGD (and other approaches).
Increasing it further does not help as Figure 7 suggests. It is not the case for NovoGrad because of its gradient normalization. However, what the authors don't show us is a different experimental setup with some trivial objective function where it would take NovoGrad an enormous amount of steps to converge to the optimum IF the algorithm is initialized far enough from the optimum. Since NovoGrad is based on normalized gradients, its steps are pretty much just signs of the original gradient multiplied by learning rate, especially in 2D. Thus, with some initial learning rate of 0.1 and when initialized in say (1, 10^10), it would take NovoGrad about 10^9 steps to converge independently on the scale of the objective function. This is not the case for SGD whose gradients are not normalized. This is actually a strong counterargument to the main claim of the paper that "NovoGrad is robust to the choice of learning rate and weight initialization". This argument in the paper is based solely on the results of the toy problem where other methods used large learning rates.

The paper includes a set of experiments containing different methods and their hyperparameters.
When the authors use some cosine function to schedule learning rate in Table 1, they do it for AdamW and NovoGrad but they use polynomial schedule for SGD. Why? Weight decay and learning rate values for SGD and NovoGrad are very different in the same table, why? Do the authors explain why this difference may happen, they don't. I believe that there is a better answer than just "this is the result of hyperparameter tuning".

The authors note that when beta=0, NovoGrad becomes layer-wise NGD with decoupled weight decay. They suggest beta to be in [0.2, 0.5] and I am wondering which beta were used in different experiment (the default suggested value is not given).

The experimental results suggest that NovoGrad performs slightly better than SGD with momentum but it has more hyperparameters and we don't know whether the results are due to different computation efforts in hyperparameter tuning. Figure 5 compares Adam and Novograd on WikiTex-103 and shows that Adam converges faster in terms training perplexity. However, Adam's test perplexity is worse than the one of NovoGrad. Interestingly, NovoGrad's test perplexity is better than its training perplexity especially in the beginning. It seems that NovoGrad is good here because it does not converge well, i.e., the results are likely due to regularization which is problematic in the original Adam.

**Experience Assessment:**

I have published one or two papers in this area.

**Review Assessment: Checking Correctness Of Derivations And Theory:**

I carefully checked the derivations and theory.

**Review Assessment: Checking Correctness Of Experiments:**

I carefully checked the experiments.

**Review Assessment: Thoroughness In Paper Reading:**

I read the paper thoroughly.

---

> ### Author Response · Authors · 2019-11-15
> **Response to reviewer 1**
>
> Thank you for the review, and especially for questions Q1 and Q3 below. Thinking how to answer them was very helpful!
>
> Q1: "It would take NovoGrad an enormous amount of steps to converge to the optimum if the algorithm is initialized far enough from the optimum. Since NovoGrad is based on normalized gradients, its steps are pretty much just signs of the original gradient multiplied by learning rate, especially in 2D. Thus, with some initial learning rate of 0.1 and when initialized in say (1, 10^10), it would take NovoGrad about 10^9 steps to converge independently on the scale of the objective function. This is not the case for SGD whose gradients are not normalized."
> A1: Even for vanilla SGD the convergence for deep linear networks is guaranteed only when network is initialized in such way that 1) objective function is close to global minimum (Bartlett et al.,  "  Gradient descent with identity initialization..., 2018), or 2) when the init point is close to the target solution in certain sense (Aurora et al, "A Convergence Analysis of Gradient Descent for Deep Linear Networks" 2018 ).
> Let's take SGD as example. For any fixed learning rate and number of steps N, I can choose the initial point close enough to $(0,0)$ such that $L(w(T)) > 1/2$ at final point $w(T)$ because the gradients near $(0,0)$ almost vanish. Algorithms with Normalized Gradients (e.g. NovoGrad) escape this saddle point without any difficulty.
>
> Q2:"Why authors use cosine function to schedule learning rate for AdamW and NovoGrad but they use polynomial schedule for SGD"?
> A2: We got the best results for ResNet-50 SGD baseline with polynomial decay. For AdamW we used cosine following the suggestions of authors of AdamW. For NovoGrad we used both cosine and poly decay, but accuracy of the model trained with cosine decay was marginally better.
>
> Q3: "Why weight decay and learning rate values for SGD and NovoGrad are different"?
> A3: Assume for simplicity that $\beta_1=\beta_2=0$ in 1st and 2nd moments (no averaging). Both Adam and NovoGrad use normalized gradients to compute the update step.  In the initial phase, normalized gradients have larger magnitudes than unnormalized gradients used by SGD. For Adam, safe learning rates are much smaller than those of SGD as the gradient elements are divided by their magnitudes and are +/-1. For NovoGrad, safe learning rates are somewhere between those of SGD and Adam as the gradients are normalized by per-layer gradient norm. Per-layer grad norms are strictly bigger than norm of gradient components used in Adam normalization.
>
> Q4: "Which beta2 were used in different experiments?"
> A4: The default value for $\beta_2=0.25$, which we used in all our experiments (ASR, LM, NMT), except ResNet-50. For ResNet-50 we used the earlier version of the code with default $\beta_2=0.98$.
>
> Q5: "Fig. 5 compares Adam and NovoGrad on WikiTex-103 and shows that Adam converges faster in terms training perplexity. However, Adam's test perplexity is worse than the one of NovoGrad. Interestingly, NovoGrad's test perplexity is better than its training perplexity especially in the beginning."
> A5: Training perplexity depicted in the figure with WikiText-103 learning curve is with dropout turned-on, validation perplexity is with dropout turned-off which makes it a priori lower.

---

### Decision · Program_Chairs · 2019-12-19

**Decision:**

Reject

**Comment:**

The paper presented an adaptive stochastic gradient descent method with layer-wise normalization and decoupled weight decay and justified it on a variety of tasks. The main concern for this paper is the novelty is not sufficient. The method is a combination of LARS and AdamW with slight modifications. Although the paper has good empirically evaluations, theoretical convergence proof would make the paper more convincing.